# AT-TSVM: Improving Transmembrane Protein Inter-Helical Residue Contact Prediction Using Active Transfer Transductive Support Vector Machines

**DOI:** 10.3390/ijms262210972

**Published:** 2025-11-12

**Authors:** Bander Almalki, Aman Sawhney, Li Liao

**Affiliations:** Department of Computer and Information Sciences, University of Delaware, Smith Hall, 18 Amstel Avenue, Newark, DE 19716, USA; alathwny@udel.edu (B.A.); asawhney@udel.edu (A.S.)

**Keywords:** bioinformatics, transmembrane protein, residues contact, transductive learning, transfer learning, contact map

## Abstract

Alpha helical transmembrane proteins are a specific type of membrane proteins that consist of helices spanning the entire cell membrane. They make up almost a third of all transmembrane (TM) proteins and play a significant role in various cellular activities. The structural prediction of these proteins is crucial in understanding how they behave inside the cell and thus in identifying their functions. Despite their importance, only a small portion of TM proteins have had their structure determined experimentally. Inter-helical residue contact is one of the most successful computational approaches for reducing the TM protein fold search space and generating an acceptable 3D structure. Most current TM protein residue contact predictors use features extracted only from protein sequences to predict residue contacts. However, these features alone deliver a low-accuracy contact map and, as a result, a poor 3D structure. Although there are models that explore leveraging features extracted from protein 3D structures in order to produce a better representative contact model, such an approach remains theoretical, assuming the structure features are available, whereas in reality they are only available in the training data, but not in the test data, whose structure is what needs to be predicted. This presents a brand new transfer learning paradigm: training examples contain two sets of features, but test examples contain only one set of the less informative features. In this work, we propose a novel approach that can train a model with training examples that contain both sequence features and atomic features and apply the model on the test data that contain only sequence features but not atomic features, while still improving contact prediction rather than using sequence features alone. Specifically, our method, AT-TSVM, employs Active Transfer for Transductive Support Vector Machines, which is augmented with transfer, active learning and conventional transductive learning to enhance contact prediction accuracy. Results from a benchmark dataset show that our method can boost contact prediction accuracy by an average of 5 to 6% over the inductive classifier and 2.5 to 4% over the transductive classifier.

## 1. Introduction

Proteins are crucial biomolecules that perform a diverse range of functions in living organisms. A specific and important type of proteins is transmembrane (TM) proteins, which span the lipid bilayer and play essential roles in various cellular functions such as signal transduction, ion transport, and membrane fusion [1]. Alpha-helical transmembrane proteins, in particular, are a common type of transmembrane protein with helices spanning the lipid bilayer [2]. Despite its critical role in various cellular activities, only a limited number of TM proteins have known structures [3]. In a recent study, among all proteins deposited in the Protein Data Bank (PDB), less than 0.7% are TM proteins [4]. Determining the TM protein structure can help to understand their function and design therapeutic drugs [5]. To better understand TM proteins’ functions, researchers have developed various computational tools to predict their structures. One of the most successful approaches focuses on predicting which amino acid residues within the transmembrane domain are in contact [6]. This approach relies on the theory that a satisfactory 3D structure of TM proteins can be built considering only residue-residue contacts [7]. As a result, several methods have been proposed for predicting the contact status of residues in proteins, including co-evolution analysis [8,9,10], the utilization of 2D topological information [11], machine learning approaches [12], and recently various deep learning architectures [13].

Most of these techniques try to predict the residue–residue contact depending only on the primary sequences of proteins (ab initio). However, with the growing size of the protein structural databases, such as Protein Data Bank (PDB) and the increasing number of 3D structures of transmembrane proteins, new approaches have become feasible, such as template-based modeling (TBM), also known as homology modeling. In TBM, the target protein whose structure is to be predicted is compared to a set of experimentally determined protein structures known as templates. However, template proteins are not guaranteed to be found, especially for proteins with low sequence similarity [14]. Another recent and highly accurate technique for predicting residue contacts in proteins is using specific atomic features extracted from protein structures [15]. Although this approach can deliver unprecedented accuracy for contact prediction, it only serves as a theoretical upper bound regarding how much information the atomic features may contribute to contact prediction, but it does not apply to real-life scenarios where the structural information is unavailable in the test data to which the trained model is deployed.

This presents a brand new transfer learning paradigm: training examples contain two sets of features—where one set is more informative than the other—but test examples contain only one set of less informative features. Currently, to our best knowledge, all machine learning models require the train and test examples to be represented in the same way: $y=f(\lambda ,\vec{x})$, where an example $\vec{x}$, regardless train or test, is represented as vector of features, *f* is the model that learns how the example’s label *y* is a function of the feature vector, and λ collectively represents model parameters, such as the weights of neurons in a neural network. In the literature, transfer learning typically refers to training the model with examples from a relevant data source, say D’, which is different from the main data source D—note that these examples from D’ still need to be represented as feature vectors of the same dimension as those from D. When the data source D contains a small number of training examples, the model is not trained well. So, transfer learning trains the model with examples (there are more) from D’ first, not because this trained model will perform any better on examples from D but because the model parameters λ thus trained with D’ are typically better off than random initialization, so when the model is further trained with examples from D, it will perform better than when it is only trained with examples in D from scratch. Our situation is quite different: we have one data source from which the training and test examples are drawn in an independently identically distributed manner, but the training examples have some extra, more informative features. How can we utilize these extra, more informative features to train a model so it can still work on the test examples (with a lack of these extra features) and perform better—via some novel form of transfer learning—than not using these extra informative features at all?

In this work, to tackle this challenge, we propose a novel approach based on transductive learning with the advantage of having training examples and test examples simultaneously, which allows us to utilize the training examples’ known structure to extract the atomic features and learn from these features, then transfer this knowledge to the test examples to improve the prediction accuracy of their inter-helical residue–residue contact. The results from a benchmark dataset show that our proposed method can improve the contact prediction accuracy in the TM proteins by an average of 5 to 6% over the inductive classifier and 3 to 4% over the transductive classifier by transferring the knowledge learned from atomic features of the training examples.

## 2. Results and Discussion

To measure the impact of our approach of transferring information from the atomic features during the model training phase to the test data, and its role in improving TM protein residue contact prediction, we first compare inductive SVM with TSVM. Then, to make sure that the performance improvement of our model is not a result of using transductive learning only [16], we compare TSVM with our proposed method, AT-TSVM. The performance is evaluated by common metrics defined below and is compared between these classifiers.(1)$$Precision=\frac{TP}{TP+FP}$$(2)$$Recall=\frac{TP}{TP+FN}$$(3)$$F1\;score=2*\frac{Precision\;*\;Recall}{Precision+Recall}=\frac{2TP}{2TP+FP+FN}$$
where *TP*, *FP*, *TN*, and *FN* stand for true positive, false positive, true negative, and false negative, respectively, when a predicted label (contact vs. no-contact) is compared with the ground truth label. The ROC score (receiver operator characteristic) is the area under the curve that plots the true positive rate versus the false positive rate as the threshold on the prediction score slides from most stringent (where every test example is predicted negative) to most relaxed (where every test example is predicted positive), thus offering a holistic measurement of the classifier’s predictive power in differentiating positive examples from negative examples and not being tied to a particular cutoff on the prediction score. ROC score ranges from 0 to 1, with 1 being achieved by a perfect classifier and 0.5 by a random classifier. In practice, for a real application scenario, though, a threshold still has to be set in order to give a binary classification. Given a classifier, there is inherent tension between precision and recall, namely, a more stringent threshold tends to yield fewer false positives, and thus higher precision, but may have more false negatives, thus lower recall. The F1 score is often preferred as it offers a balanced assessment combining both precision and recall.

Table 1 shows that TSVM is better than inductive SVM by an average of 2 to 3% in predicting TM protein contact/non-contact residues (Figure 1a). On the other hand, our proposed model, AT-TSVM, can deliver a much better residue contact performance by an average of 2.5 to 4% at the model peak (Figure 1b). However, the model can suffer from performance degradation after reaching the peak. The performance degradation could be a sign of overfitting. Therefore, to avoid overfitting, two methods are used: (1) a validation set (Figure 1c) and (2) active learning (Figure 1d). Figure 1c shows the use of different sizes of validation sets to capture the peak. In all experiments, we found that a 20 validation set is enough to capture the peak and stop the training early before overtraining. Delayed Active learning can also be used to guide the model during uncertainty and maintain the good accuracy of the model. In our case, AT-TSVM uses active learning to query the oracle for the uncertain points only. The results demonstrate that our proposed method effectively captures the model’s peak, resulting in an average contact accuracy improvement of 2.5 to 4% compared to TSVM and a further 5 to 6% improvement compared to inductive SVM. It can be seen that the validation set improvement over active learning is minimal, with an increase of only a small fraction of a percentage point. Nonetheless, in scenarios where the validation set constitutes a significant proportion—such as 20% or more—of the data, particularly in small datasets, active learning becomes a valuable option to capture the peak using only a very limited number of queries.

In comparison, DeepHelicon, from which the benchmark data sets are adopted in this study, is a top method that outperforms other state-of-the-art methods on contact prediction using sequential features only, including Germin [17], MetaPSICOV [18], and DeepMetaPSICOV [19], and reports F1 scores ranging from 0.1206 to 0.5101 under various settings. Our performance (AT-TSVM), shown in the last column of Table 1, has F1 scores ranging between 0.76 and 0.78. Note that DeepHelicon trains and tests residues as arranged by sequences. In our work, residue pairs are extracted from the sequences and are trained and tested individually. Although such an arrangement loses the contextual info that may be conducive, it offers flexibility suitable for the transductive learning framework. So, it may not be directly comparable due to different settings.

## 3. Materials and Methods

### 3.1. Dataset

The dataset used in this study is adopted from DeepHelicon [17] and consists of 222 α-helical TM proteins with a resolution better than 3.5 Å, with a number of TM helices ranging from 2 to 17. According to DeepHelicon, a contact point is defined as 2 residues that are separated by a minimum of 5 residues in sequence and for which the minimum distance between any pair of their heavy atoms measures less than 5.5 Å in the ground truth structure from the Protein Data Bank. In this study, we randomly sample contact and non-contact residues from this dataset to train and test our proposed method.

#### 3.1.1. Sequence Features

Among all sequence features extracted from protein sequences to predict residue contacts in proteins, co-evolutionary features have been shown to be the most informative [20]. To extract the co-evolutionary features, we follow the idea of [12], where four direct coupling analysis models (CCmpred [8], EVfold [10], plmDCA [21], and Gaussian DCA [22]) are used to calculate the coupling scores. In [12], for a residue pair at position (i,j) of a sequence alignment, the residues (i+x,j+y),(i+x,j−y),(i−x,j−y), and (i−x,j+y) are considered, where (x,y)∈(0,0),(0,1),(0,3),(0,4),(1,0),(3,0),(3,4),(4,0),(4,3),(4,4), resulting in a 100-feature vector. In addition, a moving window of length three is applied to this vector, resulting in a vector of length 300. To reduce the high-dimensional feature space, we first applied Principal Component Analysis (PCA) to the 300 features. Then, we ranked these PCAs by their contact predictive power instead of variance. $PCA_{Best1}>PCA_{Best2}>$… $PCA_{Best300}$. After that, the top 10 PCAs by prediction power were chosen (Figure 2).

#### 3.1.2. Atomic Features

In [15] it is shown that atomic features extracted from protein structures can deliver a very high-accuracy prediction. We follow the method presented in [15] to extract the following atomic features from the protein structure: mean distance between atoms, relative residue angle, $C_{\alpha }$ atom distance, deviation of distances between atoms, and inter-helical angle. Among these atomic features, the most informative is the mean distance, as reported in [14]. In [15], for a residue pair at position (i,j), the neighboring positions (i,j)(i+x,j+y) are used, where (x,y)∈{(1,1),(−1,−1),(1,−1),(−1,1),(0,1),(0,−1),(1,0),(−1,0)}. This 3×3 window, excluding the center, has eight neighbor residue pairs. Five features per residue pair are used, resulting in 8×5=40 features. Again, like in sequential features, we apply PCA to reduce the high-dimensional feature space to only two features (Figure 2).

#### 3.1.3. Feature Fusion

Feature fusion refers to combining various feature representations from data to produce more informative features [23]. In protein residue contact prediction, atomic features have been shown to be very informative (i.e., leading to high accuracy prediction) as compared to the sequence features. However, since these atomic features are extracted from protein structures, they cannot be used at the model testing phase and hence can only serve as a theoretical upper bound. Therefore, we propose a novel approach that fuses features extracted from proteins’ sequences and structures during training but only uses protein sequences during testing. In this approach, during training, the model has access to both the protein’s sequence features and its corresponding 3D structure to extract the atomic features. During testing, however, the model has access only to the sequence features and tries to infer the missing atomic features. This is achieved by using a learning technique called transductive learning, Transductive Support Vector Machines to be specific, enabled by novel transfer and active learning techniques.

### 3.2. Method

#### 3.2.1. Transductive Support Vector Machines

Transductive learning is a form of learning used to infer the values of unlabeled data in a given set based on knowledge of the seen data within that set [24]. In contrast to the traditional supervised learning approach, which uses only labeled training data to train a classifier and then the trained classifier to classify unlabeled data, transductive learning uses both labeled and unlabeled data to make a more accurate prediction.

Transductive Support Vector Machines (TSVMs) are a specific type of Support Vector Machines that use transductive learning to make predictions from both labeled and unlabeled data. An early implementation of TSVM is the $SVM_{light}$ [25] algorithm, which minimizes misclassifications by utilizing unlabeled test examples during training (Figure 3). The goal of the transductive learner *L* in this algorithm is to find a hypothesis from the hypothesis space *H*, such that the misclassification is reduced.(4)$$R\left(L\right)=\int \frac{1}{k}\sum_{i=1}^{k}\theta \left(h_{L}\left({\vec{x}}_{i}^{*}\right),y_{i}^{*}\right)\;dP\left({\vec{x}}_{1},y_{1}\right)\;dP\left({\vec{x}}_{k}^{*},y_{k}^{*}\right).$$
where *k* is the number of test examples and $h_{L}$ is the Hinge Loss. θ(a,b)=0 if a=b and zero otherwise. Then, the TSVM objective function is used to minimize the following function:(5)$$\frac{1}{2}{∥w∥}^{2}+C\sum_{i=1}^{n}\xi_{i}+C_{-}^{*}\sum_{j:y_{j}^{*}=-1}\xi_{j}^{*}+C_{+}^{*}\sum_{j:y_{j}^{*}=1}\xi_{j}^{*}$$
where $C,C_{+}^{*}$, and $C_{-}^{*}$ are hyperparameters for the misclassification penalty of labeled examples, predicted positive unlabeled examples, and predicted negative unlabeled examples, respectively. $\xi_{i}^{*}$ and $\xi_{j}^{*}$ are slack variables that specify how far the misclassified training and test examples, respectively, are from their corresponding margins. The $SVM_{light}$ implementation of TSVM consists of two nested loops. While the outer loop is used to assign labels for the unlabeled test examples, the inner loop is the core of this algorithm, where labels are swapped iteratively to find a hyperplane that can separate positive and negative data points.

We use TSVM to train our model on both the labeled training set and the unlabeled test set, ${(\vec{x}}_{1},y_{1}),\;{(\vec{x}}_{2},y_{2}),\ldots ,\;{(\vec{x}}_{n},y_{n})\;\cup {\vec{x}}_{1}^{*},{\vec{x}}_{2}^{*},\ldots .,{\vec{x}}_{k}^{*}$. However, while the training set includes the full feature set (sequence and atomics) ${\vec{x}}_{i}=s_{1},s_{2},\ldots \ldots ,s_{10},\;a_{1},a_{2}$, the test set contains only a partial feature set (sequence features only). ${\vec{x}}_{i}^{*}=s_{1},s_{2},\ldots \ldots ,s_{10}$. As a result, more preprocessing is required before model training. Notice here that the atomic features are missing from all the test examples. Therefore, traditional data imputation techniques do not apply to this situation. Figure 4a demonstrates why a simple inductive classifier fails to handle this scenario, primarily due to the absence of atomic features during the model’s testing phase. To overcome this issue, we use the positive and negative atomic feature means of the training set as a proxy to start the training process. To accomplish this, we first extract the mean of positive and negative atomic features from the training set. Then, we use a separate inductive SVM classifier trained and tested only on the sequence features to predict contact/non-contact residues. Depending on this classifier’s output, we assign the positive and negative atomic means to the test set.(6)$$a_{i}^{*}=\mu \;A|Y=E\left[A|Y\right]=\sum_{a}aP\left(A=a\right|Y)$$

The conditional mean is used to the mean of extract atomic features from training examples, where A is the atomic feature of the training set and Y is the classifier label (contact/no contact). Now the test set has a complete feature set $\;{\vec{x}}_{i}^{*}=s_{1},s_{2},\ldots \ldots ,s_{10},\;a_{1}^{*},a_{2}^{*}$, where $a_{i}^{*}$ is the positive/negative atomic mean of the test set. Figure 4b illustrates our proposed approach for addressing the absence of atomic-level features during the model’s testing phase.

#### 3.2.2. Transfer Learning

Transfer learning is a type of learning that focuses on reusing knowledge acquired while solving one task to apply it to another related task [26]. This knowledge can be used to solve a new problem, as well as improve the efficiency of the learning process. As mentioned in the Introduction, this is different from the typical transfer learning that is commonly used for pretraining the model with examples from a related but different data source so that the model parameters will be initialized better than random values. Here, in our study, there is just one data source from which the train and test examples are drawn in an independently and identically distributed (i.i.d.) manner, but the training examples have some extra, more informative features—the atomic features extracted from 3D structures determined by costly X-ray crystallography experiments. Our task is to utilize these extra, more informative features to train a model so it can still work on the test examples (lack of these extra features) and perform better than not using these extra informative features at all. So the transfer learning is new and quite different from how it is commonly used.

Transductive learning, in which the test examples participate in the model training as unlabeled data, together with the labeled training data, offers a convenient framework for our transfer learning. Specifically, we adopted a Transductive Support Vector Machine for this purpose. Test examples, lacking atomic features, cannot be put into the same vector space as the training examples that contain both sequence features and atomic features. So test examples have to be augmented with atomic features to participate in the transductive learning. This initialization can be assigned randomly or with some more refined schemes—we use an inductive SVM that is trained only on the sequential features, which are available to both training examples and test examples, and augmented test examples with the average atomic features from the training examples from the same class; that is, if a test example is predicated by the inductive SVM as positive, then it is augmented with the average atomic features from positive training examples; if a test example is predicated by the inductive SVM as negative, then it is augmented with the average atomic features from positive training examples. Once the test examples participate in transductive learning, the transfer learning iteratively updates the atomic feature assignment to the test examples to optimize the objective function. Specifically, in addition to the traditional TSVM algorithm (Algorithm 1), which only swaps the label of any eligible pairs that would decrease the objective function (Equation (Equation 5)), we update the atomic features at each iteration of the TSVM. We use the K-Nearest Neighbor (KNN) algorithm (Algorithm 2) to update the atomic features at each TSVM iteration for the swapped data, depending on their assigned labels, correspondingly. In particular, each atomic feature $a_{i}^{*}$ is updated using their k nearest positive/negative training atomic means (Figure 5).

The Euclidean distance d$\left(p,q\right)=\sqrt{\sum_{i=1}^{n}{\left(q_{i}-p_{i}\right)}^{2}}$ is used to calculate the k nearest neighbors. K is a hyperparameter that can be set by the user. We found that k = 10 to 20 produced the best performance and hence will be used. This update ensures that the atomic feature values always comply with their labels at every iteration. It is worth noting that using the absolute mean instead of KNN to update the atomic values of the test set at each iteration is not enough to deliver a good performance. A possible reason is that using the absolute mean would push the point far away from the hyperplane to its inner territory and inhibit its influence on the hyperplane. Therefore, using KNN is a crucial part of our model to boost the contact accuracy of the TM protein’s residues.
**Algorithm 1** TSVM Algorithm**Input:**$\left({\vec{x}}_{i},y_{i}\right)\;\left({\vec{x}}_{1}^{*},{\vec{x}}_{2}^{*},\ldots .,{\vec{x}}_{k}^{*}\right)$ //labeled training and unlabeled test sets, respectively.$C\;C^{*}$ // misclassification penalty for labeled training and unlabeled test examples, respectively.**Output:**$y_{1}^{*},\ldots ..y_{k}^{*}$ //predicted labels of test examples**Procedure:****while** 
$\left(\left(C_{-}^{*}<C^{*}\right)\vee \left(C_{+}^{*}<C^{*}\right)\right)$ 
**do**    // call solve-svm-qp, a standard procedure to solve quadratic programming    // for soft margin SVM [25].    $\left(\vec{w},b,\vec{\xi },{\vec{\xi }}^{*}\right)=\mathrm{solve}\text{-}\mathrm{svm}\text{-}\mathrm{qp}\left(\left[\left({\vec{x}}_{1},y_{1}\right)\ldots .\left({\vec{x}}_{n},y_{n}\right)\right],\left[\left({\vec{x}}_{1}^{*},y_{1}^{*}\right)\ldots .\left({\vec{x}}_{k}^{*},y_{k}^{*}\right)\;\right],C,C_{-}^{*},C_{+}^{*}\right)\;$    **while** $\left(\exists m,l\left(y_{m}^{*}*y_{i}^{*}<0\right)\;\&\;\left(\xi_{m}^{*}>0\right)\;\&\;\left(\xi_{i}^{*}>0\right)\;\&\;\left(\xi_{m}^{*}+\xi_{i}^{*}>2\right)\right)$ **do**        $y_{m}^{*}=-y_{m}^{*}$;        $y_{i}^{*}=-y_{i}^{*}$;        Transfer Learning ( ) // **call Algorithm 2**        Active Learning ( ) // **call Algorithm 3**    **end while****end while**

**Algorithm 2** Transfer Learning Algorithm

**Input:**
posA, NegA // The positive and negative atomic means of the train set, respectively.$a_{i}^{*},y_{i}$ // The $i^{th}$ test atomic feature and its corresponding label.*k* // A hyperparameter determined by the user represents the number of the K-Nearest Neighbors to consider.
**Output:**
$\;a_{i}^{*}$ // The updated atomic features.
**Procedure:**

*Step 1: Let P,N be lists that contain the indices of the contact and non-contact residues in the training set.*

*Step 2: Use the Euclidean distance to calculate the distance from point i to the k nearest points of $\vec{X}\left[P\right]$ and X[N] points.*

*Step 3: Update $a_{i}^{*}$*
**if** $y_{i}$ is positive (contact) **then**    $a_{i}^{*}=posA$
**else**
    $a_{i}^{*}=NegA$
**end if**
**return** $a_{i}^{*}$


**Algorithm 3** Delayed active learning

**Input:**
L,U // Labeled training set and unlabeled test set.*R*// Current TSVM iteration number.*k* // The number of test examples to label at each iteration.*S* // The iteration number to start Active Learning.
**Output:**
$y_{1}^{*},y_{2}^{*}$……, $y_{k}^{*}$ // the actual labels of the test examples.
**Procedure:**
**if**thenR>=S:    *Step 1: Use the SVM decision function $f({\vec{x}}_{i}^{*})$ to calculate how far each test data point is from its corresponding margin.*    *Step 2: Rank $f({\vec{x}}_{i}^{*})$ in descending order and choose the first k test data points to query the oracle.*    *Step 3: provide the true labels for these k points.*    *Step 4: Remove these k points from U and add them to L.*
**else**
    continue
**end if**



Note that, for our method and any other machine learning and statistical learning methods to work, the premise is that training examples and test examples are drawn in an i.i.d. manner from the same data source. Under this premise, the method should work for other transmembrane helical proteins outside the data set used in this study, as long as they belong to the same overall data source. Otherwise, the models trained on certain data cannot generally be expected to work well on data that are from a different source with a different underlying distribution. That said, our method can still be used on a different set of proteins with a different underlying distribution if that set contains examples with known residue contacts to serve as training examples. Actually, one advantage of our transfer transductive learning method is that it requires a small training set because the test example can participate in training as unlabeled data.

#### 3.2.3. Active Learning

Active learning (AL) is a machine learning approach where the algorithm actively selects and queries the most informative or uncertain instances from an unlabeled dataset to be labeled by an oracle and uses these labeled instances to update and improve its model [27]. In TSVM, the most uncertain points are usually those close to the margin [28]. Therefore, we use active learning to ask the oracle about true labels for a limited, small number of data points to prevent overtraining. However, we noticed that using the traditional AL from the beginning of the model training would cause the model to stop early before reaching the peak. Therefore, we use our modified AL algorithm (Algorithm 3) to postpone the start of the AL until the performance starts to drop. To achieve that, a validation set is used to capture the F1 peak. Then, we choose that peak as the starting point for the AL. Doing so would help the model to prevent uncertainty, reduce misclassification, avoid overfitting, and thus stop early. Using only two points at each TSVM iteration is shown to be enough to guide the model to stop before overfitting.

A few points about active learning are worth mentioning. Firstly, in real-world applications, active learning is only feasible when obtaining the ground truth for the inquiries from the learner is less costly, but it should still be allowed only for limited inquiries; otherwise, the learner can ask the oracle for answers to all test examples. Given that AlphaFold [29,30] has made strides in protein 3D structure prediction with great success, it is conceivable that the predicted structure by AlphaFold may play a role as a low-cost Oracle or surrogate to the ground truth, though using AlphaFold’s predicted structure puts the whole study under a different premise and is beyond the current scope. Secondly, when the learner inquires the oracle for an answer in a difficult case, it should not matter how the oracle obtains the answer—either from the exiting ground truth label or when the ground truth is not available and therefore has to carry out real experiments—as long as these examples can no longer participate in evaluating the test performance—that is, no cheating is allowed. This is exactly the procedure followed here; test examples participated in active learning are not used in evaluating the learner’s performance. Also, in a transductive learning mode, as adopted in this study, the test examples are involved in training and are predicted along the way; therefore, the concerns of training on the test are moot in this case. Lastly, note that the performance of active learning is no better (and actually slightly worse) than the validation set, as reported in Table 1. Here, the reason that active learning is adopted as an alternative is not because it delivers better performance—it does not—but rather because it requires much fewer test examples to ask the oracle than using a validation set, order to know when the transductive learning should stop to avoid overtraining. This is particularly more helpful when the labeled data are limited.

## 4. Conclusions

In this work, we developed a novel approach that enables transfer learning in the framework of a transductive classifier, allowing us to train on examples that have features extracted from both protein sequences and known 3D structures and make contact residue prediction for test examples that have features only the sequence, not the atomic features. Iteratively, during the training process, via a novel transfer and active learning scheme, the classifier infers the missing atomic features for these proteins that do not have a 3D structure to improve the quality of the residue contact prediction. Our proposed method demonstrates a notable enhancement in predicting contact between inter-helical residues in TM proteins. On average, it achieves a 3 to 4% improvement compared to the conventional TSVM approach and a 5 to 6% improvement compared to the inductive SVM method. This improvement stands in contrast to relying solely on features extracted from protein sequences. While this approach is designed specifically for inter-helical residue contact prediction, the innovative technique may lead to a new learning paradigm and can be adapted to other applications in which some informative features are only available in the training data. Future work will further improve the performance by integrating the transfer learning with the kernel engineering of TSVM.

## Figures and Tables

**Figure 1 ijms-26-10972-f001:**
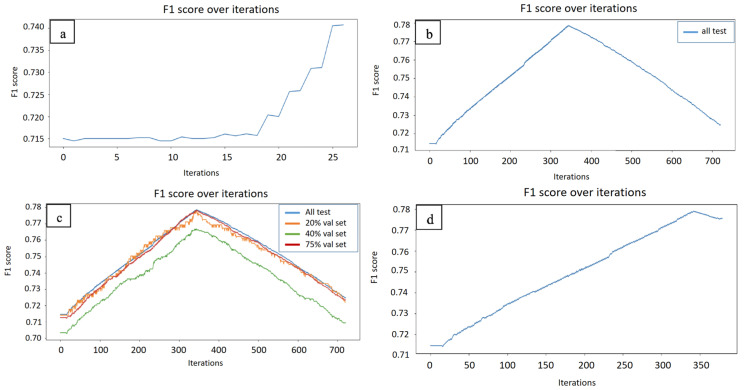
Monitoring F1 score over iterations of TSVM and AT-TSVM. (**a**) TSVM F1 score increases gradually over iterations but peaks at a lower F1 score. (**b**) AT-TSVM F1 score shows a steady increase to reach a higher F1 score before starting to drop. (**c**) F1 score shows that different-sized validation sets can capture the peak of the model. (**d**) AT-TSVM (with active learning) can capture the peak, stop early, and avoid overfitting with a very limited query.

**Figure 2 ijms-26-10972-f002:**
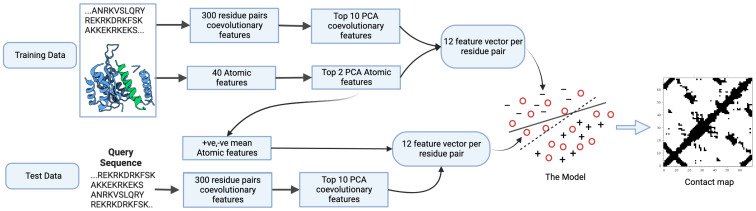
AT-TSVM pipeline. The model is trained on both sequence and atomic features. The atomic features are extracted from known protein structures. During testing, the model has access to the sequence features only and tries to infer the missing atomic features. The output of the model is a 2D contact map.

**Figure 3 ijms-26-10972-f003:**
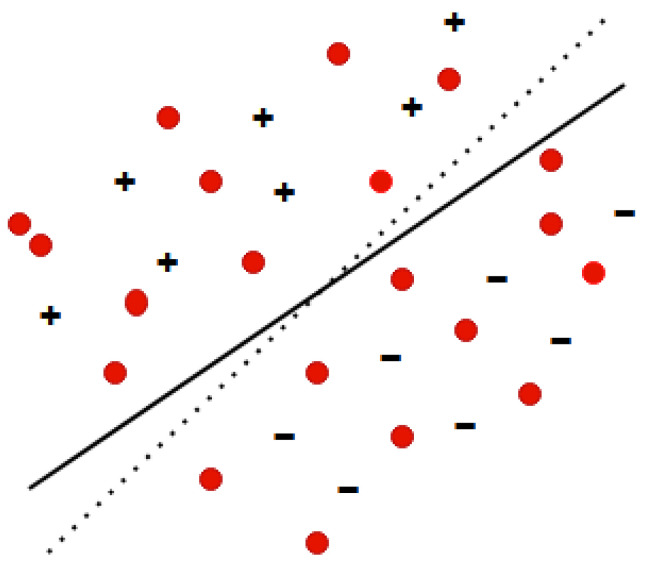
Inductive SVM using the labeled data (marked as + and −) to learn a separating hyperplane (solid line) versus Transductive SVM using the unlabeled data in addition (marked as a red dot) to learn a separating hyperplane (dashed line).

**Figure 4 ijms-26-10972-f004:**
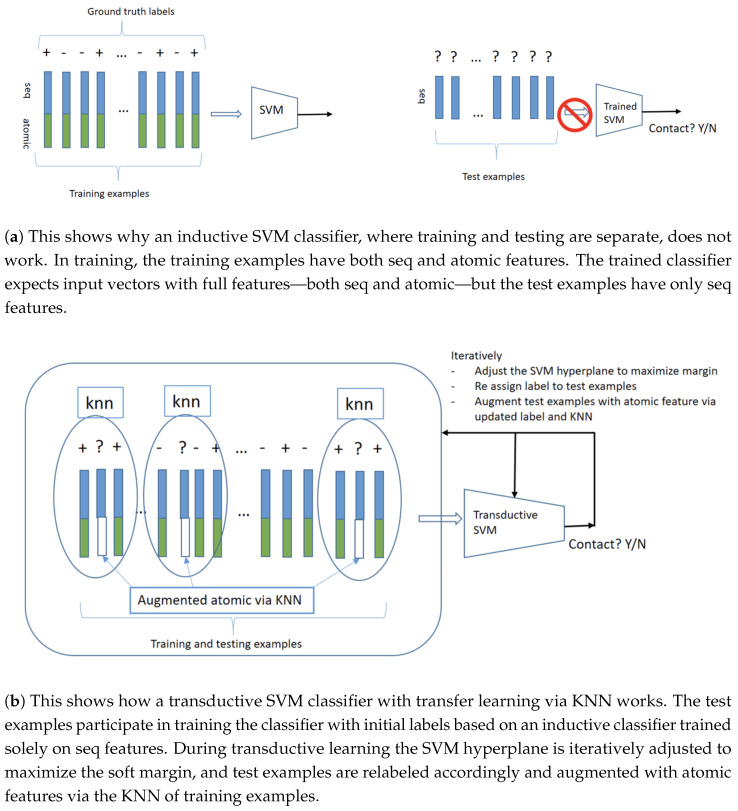
Overview of the limitations of inductive classification and the proposed transductive approach. (**a**) An inductive SVM fails when atomic features are missing at the test time, as it relies on full feature vectors seen during training. (**b**) A transductive SVM with transfer learning via KNN overcomes this limitation by incorporating test examples during training, initializing labels using a sequence-only classifier, and iteratively refining labels and imputing atomic features through nearest neighbors.

**Figure 5 ijms-26-10972-f005:**
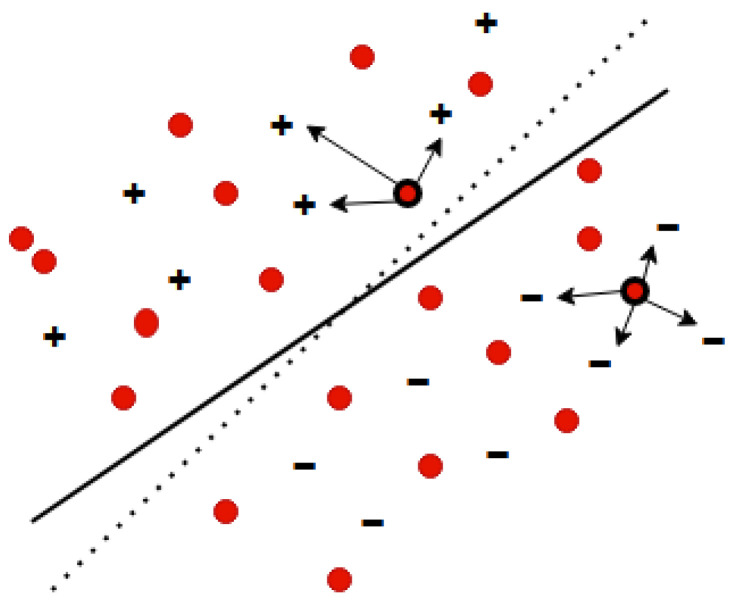
Updating Test Examples’ Atomic Features Using KNN. Red points represent test examples that miss atomic features’ values. To compensate for these missing values, the atomic features’ values of the K nearest neighbors are used. The + and − signs stand for positive and negative training examples. The dashed line is the separating hyperplane learned by inductive SVM, and solid line is the separating hyperplane learned by tranductive SVM. The arrow points to the neighboring training examples identified by KNN.

**Table 1 ijms-26-10972-t001:** Inductive SVM and TSVM on sequence features VS our model (AT-TSVM) prediction performance.

Trains Set	Test Set	SVM Average Scores on Seq Features	TSVM Average Scores on Sequence Features	AT-TSVM Average F1 Scores (Using Validation Set)	AT-TSVM Average F1 Score (Using Active Learning)
1000(400c,600n)	5000	precision: 0.812 ± 0.0089 Recall: 0.6478 ± 0.0251 F1: 0.7202 ± 0.0135 ROC: 0.7738 ± 0.0090	precision: 0.7649 ± 0.0074 Recall: 0.7155 ± 0.010 F1: 0.7393 ± 0.0082 ROC: 0.7844 ± 0.0065	precision: 0.8253 ± 0.010 Recall: 0.7132 ± 0.014 F1: 0.7650 ± 0.0063 ROC: 0.8061 ± 0.0047	precision: 0.8265 ± 0.0113 Recall: 0.7136 ± 0.0145 F1: 0.7661 ± 0.0071 ROC: 0.8068 ± 0.0053
2000(800c,1200n)	5000	precision: 0.8213 ± 0.0151 Recall: 0.6364 ± 0.0132 F1: 0.7179 ± 0.0072 ROC: 0.7717 ±0.0051	precision: 0.7867 ± 0.0204 Recall: 0.6823 ± 0.0128 F1: 0.7333 ± 0.0056 ROC: 0.7802 ± 0.0045	precision: 0.8354 ± 0.0158 Recall: 0.7051 ± 0.0119 F1: 0.7696 ± 0.0073 ROC: 0.8076 ± 0.0056	precision: 0.8341 ± 0.0180 Recall: 0.7077 ± 0.0127 F1: 0.76617 ± 0.0055 ROC: 0.8088 ± 0.0068
1500(600c,900n)	5000	precision: 0.8221 ± 0.015 Recall: 0.6376 ± 0.0302 F1: 0.7206 ± 0.0163 ROC: 0.7739 ± 0.0105	precision: 0.7744 ± 0.0078 Recall: 0.707 ± 0.0147 F1: 0.7406 ± 0.0080 ROC: 0.7852 ± 0.0061	precision: 0.8296 ± 0.0188 Recall: 0.7245 ± 0.0083 F1: 0.7720 ± 0.0076 ROC: 0.8102 ± 0.0070	precision: 0.8330 ± 0.0189 Recall: 0.7207 ± 0.0142 F1: 0.7702 ± 0.0090 ROC: 0.8110 ± 0.0073
2500(1000c,1500n)	10,000	precision: 0.8153 ± 0.0090 Recall: 0.6436 ± 0.0164 F1: 0.7218 ± 0.0067 ROC: 0.7740 ± 0.0060	precision: 0.7840 ± 0.0150 Recall: 0.6951 ± 0.0133 F1: 0.7351 ± 0.0023 ROC: 0.7822 ± 0.0048	precision: 0.8232 ± 0.0090 Recall: 0.7142 ± 0.0128 F1: 0.7670 ± 0.0049 ROC: 0.8067 ± 0.0053	precision: 0.8236 ± 0.0088 Recall: 0.7141 ± 0.0136 F1: 0.7668 ± 0.0050 ROC: 0.8076 ± 0.0045
3000(1200c,1800n)	10,000	precision: 0.7912 ± 0.0642 Recall: 0.6880 ± 0.0876 F1: 0.7201 ± 0.0019 ROC: 0.7727 ± 0.0068	precision: 0.7392 ± 0.0423 Recall: 0.7529 ± 0.0495 F1: 0.7397 ± 0.0040 ROC: 0.7841 ± 0.0048	precision: 0.8002 ± 0.0523 Recall: 0.7756 ± 0.0614 F1: 0.7731 ± 0.0040 ROC: 0.8112 ± 0.006	precision: 0.8002 ± 0.0524 Recall: 0.74484 ± 0.0435 F1: 0.7732 ± 0.0043 ROC: 0.8112 ± 0.0065
4000(1600c,1400n)	10,000	precision: 0.7818 ± 0.0055 Recall: 0.6941 ± 0.0080 F1: 0.7360 ± 0.0035 ROC: 0.7830 ± 0.0026	precision: 0.7344 ± 0.0030 Recall: 0.7557 ± 0.0060 F1: 0.7454 ± 0.0030 ROC: 0.7865 ± 0.0024	precision: 0.8012 ± 0.0092 Recall: 0.7686 ± 0.0138 F1: 0.7845 ± 0.0064 ROC: 0.8207 ± 0.0052	precision: 0.7988 ± 0.0066 Recall: 0.7666 ± 0.0142 F1: 0.7842 ± 0.0064 ROC: 0.8205 ± 0.0051

## Data Availability

AT-TSVM model codes and predicted contact maps used in this article are available at https://github.com/Bander-Almalki/AT-TSVM/ (accessed on 9 November 2025).

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
