# Peer review of "AT-TSVM: Improving Transmembrane Protein Inter-Helical Residue Contact Prediction Using Active Transfer Transductive Support Vector Machines"

_ijms, 2025, doi:10.3390/ijms262210972_

Round 1

Reviewer 1 Report

Comments and Suggestions for Authors

In this manuscript, the authors describe their method AT-TSVM for predicting inter-helical residue contacts of TM proteins. They claim that by using transfer learning and active learning combining with TSVM, prediction accuracy in terms of F1 score can be improved. Although this paper introduces a novel approach for TM protein contact prediction and has some interesting results, there are major concerns that need to be addressed about the results and methods.

 Major comments:

  1. Overall this work lacks suitable baseline studies and only compares within the SVM related approaches. For instance, how does the AT-TSVM approaches here compare to the DeepHelicon method where the authors adopted the data and feature extraction methods from? Additionally, deep learning based protein structure prediction methods have made significant progress in the past few years, while the authors only cited 2 papers published after 2021. How does AlphaFold/ESMFold perform on the specific test set that the authors used? Without more baseline studies, it is hard to gauge where the method stands currently. What is worth noting is that the method does not necessarily have to outperform state-of-the-art methods in terms of accuracy, as it can have other advantages such as being faster.
  2. The premise of using active learning in this setting is questionable. Active learning is usually used in cases where the ground truth is easy to obtain (while maybe slow or not scalable). In this case, no perfect oracle exists. The authors (from my understanding) seem to just use the ground truth label of the test set as oracle, which leads to the concern of training on the testing data, and these labels are generally not available in real-world scenarios. Despite this, active learning only barely improved (and in some cases degraded) accuracy. I would suggest either use some other prediction oracles such as AlphaFold, or simply removing the active learning component from the method/paper.
  3. The results/discussion sections of the paper are currently very brief. The authors should expand with more discussions on the interpretation of the results.

Minor comments:

  1. Table 1 in this format is very hard to read. Authors should bold the best-performing metrics or try to convert it to a plot.
  2. 11-3.13 should also belong to methods.
  3. AT-TSVM was not fully spelled out in the paper.
  4. “A” does not seem to be defined in algorithm 2 “if yi is positive (contact) then ai∗ = A”

Author Response

Reviewer#1

In this manuscript, the authors describe their method AT-TSVM for predicting inter-helical residue contacts of TM proteins. They claim that by using transfer learning and active learning combining with TSVM, prediction accuracy in terms of F1 score can be improved. Although this paper introduces a novel approach for TM protein contact prediction and has some interesting results, there are major concerns that need to be addressed about the results and methods.

 Major comments:

1. Overall this work lacks suitable baseline studies and only compares within the SVM related approaches. For instance, how does the AT-TSVM approaches here compare to the DeepHelicon method where the authors adopted the data and feature extraction methods from? Additionally, deep learning based protein structure prediction methods have made significant progress in the past few years, while the authors only cited 2 papers published after 2021. How does AlphaFold/ESMFold perform on the specific test set that the authors used? Without more baseline studies, it is hard to gauge where the method stands currently. What is worth noting is that the method does not necessarily have to outperform state-of-the-art methods in terms of accuracy, as it can have other advantages such as being faster.

Response: Regarding comparison to other state-of-the-art methods, we added the following in the revision.

“As comparison, DeepHelicon, from which the benchmark data sets are adopted in this study, is a top method that outperforms other state-of-the-art methods on contract prediction using sequential features only, including Gremlin [17], MetaPSICOV [18], and DeepMetaPSICOV [19], and reports F1 scores range from 0.1206 to 0.5101 under various settings. Our performance (AT-TSVM), shown in the last column of table 1, has F1 scores ranging between 0.76 to 0.78. Note that DeepHelicon trains and tests residues as arranged by sequences. In our work, residues pairs are extracted out of the sequences, and are trained and tested individually. Although such arrangement lost the contextual info which may be conducive, it offers flexibility suitable for the transductive learning framework. So, it may not be not directly comparable, due to different settings.”

AlphaFold/ESMFold are full-fledged methods for 3D structure prediction as a whole, so we didn’t consider these methods in the same category to compare with. But it is very likely that our more customized method takes less resources to train. In revision, we cited AlphaFold/AlphaFold3 in the context of active learning by potentially using it as a low cost Oracle or surrogate to the ground truth.

The premise of using active learning in this setting is questionable. Active learning is usually used in cases where the ground truth is easy to obtain (while maybe slow or not scalable). In this case, no perfect oracle exists. The authors (from my understanding) seem to just use the ground truth label of the test set as oracle, which leads to the concern of training on the testing data, and these labels are generally not available in real-world scenarios. Despite this, active learning only barely improved (and in some cases degraded) accuracy. I would suggest either use some other prediction oracles such as AlphaFold, or simply removing the active learning component from the method/paper.

Response: We totally agree with the reviewer that active learning is more practical when getting the ground truth for the inquiries from the learner is less costly – though still should be allowed for limited inquiries, otherwise we can just ask the oracle for all. There are a couple of key differences worth classifying.

First, when we inquire the oracle of answer for a tough case, it shouldn’t matter how the oracle gets the answer (either from the exiting ground truth label, or when ground truth not available and therefore has to do real experiments), as long as these examples can no longer participate in evaluating the test performance, namely no cheating. This is exactly we followed.

Second, we are in a transductive learning mode, which means that the test examples are involved in training and get predicted along the way. Therefore, the concerns of training on the testing is mooted in our case.

The reviewer is right that the performance of Active learning is no better (and actually slightly worse) than the validation-set. The reason we include active learning as an alternative is not that it delivers better performance – it does not -- but rather that it requires much fewer test examples to inquire the oracle than using a validation set, in order to know when the transducive learning should stop to avoid over training. In the revision, words to this effect is added to clarify. And we hope that the reviewer would agree to let us keep the active learning part in the manuscript.

2. The results/discussion sections of the paper are currently very brief. The authors should expand with more discussions on the interpretation of the results.

Response: Thanks for the suggestion. Words reflecting our responses to the reviewers comments are added, and the Results/discussions section is substantially expanded in the revised manuscript.

Minor comments:

1. Table 1 in this format is very hard to read. Authors should bold the best-performing metrics or try to convert it to a plot.

Response: we count on the production team from the journal to address the format to turn table 1 from sideway to upright, should the manuscript be kindly accepted for publication. We prefer the table to a plot, as it lists actual values so the reader does not have to “guess” from a plot.

2. 11-3.13 should also belong to methods.

Response: The reviewer has a good point here. Indeed, subsections 3.1.1 to 3.1.3 are not about the raw data per se, but rather about extracting features from the raw data, therefore conceptually belong to the method (sub-)section. The reason we put them under “Data” is because they are preprocessing the data to feed into the computation pipeline, and the computation pipeline is the main focus for “Method”.  We prefer this more balanced arrangement, and hope the reviewer will agree to let it stay.

3. AT-TSVM was not fully spelled out in the paper.

Response: It is now fully spelled out the abstract.

4. “A” does not seem to be defined in algorithm 2 “if yi is positive (contact) then ai∗ = A”

Response: It should be posA. The error is fixed.

Reviewer 2 Report

Comments and Suggestions for Authors

2.1

The author relies solely on the F1 score as the evaluation metric in the manuscript.

Although Precision, Recall, and ROC are listed in the table,

no description is provided regarding the specific calculation methods for these three metrics.

In terms of Precision, AT-TSVM shows only marginal improvement over SVM.

The advantages of the proposed method are not as significant as claimed.

2.2

In Algorithm 1, there is no explanation regarding qp.

2.3

During the testing phase, since the input consists only of sequence information, the author substitutes the actual atomic feature with "the positive and negative atomic feature means" (calculated from the training set) as input. Regarding this approach:

01.Lack of Distribution Guarantee: The means of the testing set cannot be guaranteed to be consistent with those of the training set.

02.Fundamental Statistical Flaw: Even if point 1 were not an issue, it is evident that the mean value of a dataset cannot represent the properties of individual samples. Given that the atomic feature is a crucial part of the network input and significantly impacts the classification results (if its impact were small, using only the positive and negative feature means as input for all samples during the training phase would have sufficed).

Therefore, the author's network design incorporates a significant flaw.

Comments on the Quality of English Language

The author's entire manuscript is riddled with formatting and word errors, with only a portion listed below. These issues reflect a lack of seriousness during the manuscript preparation stage.

line 75 "model,AT-TSVM", should be "model, AT-TSVM,"

line 79 "Figure Figure 1.c" should be "Figure 1.c"

line 151 "zero otherwise Then," should be "zero otherwise. Then,"

line 196 "The Euclidian distanced(p,q)" should be "The Euclidian distance d(p,q)"

line 177 "Figure.3(b) llustrates" should be "Figure.3(b) illustrates"

Author Response

Reviewer#2

The author relies solely on the F1 score as the evaluation metric in the manuscript. Although Precision, Recall, and ROC are listed in the table, no description is provided regarding the specific calculation methods for these three metrics.

 Response: The focus on F1 score is due to its offering a balanced way in assessing the performance, encompassing both precision and recall. ROC has the advantage of not relying on the cutoff for the prediction score to assess the performance, for more theoretic purpose. Since eventually a binary classifier has to give a yes/no prediction, metrics like precision, recall, and hence F1 score, are more practical. A whole paragraph of description for various metrics is now provided in the revision.

In terms of Precision, AT-TSVM shows only marginal improvement over SVM. The advantages of the proposed method are not as significant as claimed.

 Response: One average, AT-SVM achieves a 3 to 4% improvement over TSVM and a 5 to 6% over inductive SVM. In the domain of work, such improvements are considered significant. Also the small standard deviations indicate statistical significance of the improvement, not due to fluctuation of random train/test split.

2.2

In Algorithm 1, there is no explanation regarding qp.

 Response: qp is quadratic programming, a standard procedure for solving soft-margin SVM adopted from Reference [25]. The issue is fixed in the revision.

2.3

During the testing phase, since the input consists only of sequence information, the author substitutes the actual atomic feature with "the positive and negative atomic feature means" (calculated from the training set) as input. Regarding this approach:

01.Lack of Distribution Guarantee: The means of the testing set cannot be guaranteed to be consistent with those of the training set.

 Response: To train any useful model, the assumption is that training and testing examples are drawn i.i.d., from the same source. This is the foundation for any machine learning or statistical learning, and is also what our study is based up on. The reviewer is right that this assumption is not guaranteed in practice. So, when the train and test data follow somewhat different distributions, the trained model may not perform as well as expected. But this is an issue shared by all machine learning techniques based on that assumption, not an issue unique to our method.

02.Fundamental Statistical Flaw: Even if point 1 were not an issue, it is evident that the mean value of a dataset cannot represent the properties of individual samples. Given that the atomic feature is a crucial part of the network input and significantly impacts the classification results (if its impact were small, using only the positive and negative feature means as input for all samples during the training phase would have sufficed).

Therefore, the author's network design incorporates a significant flaw.

 Response: Using the mean of a dataset, though crude, is one of the most common way to profile the data, and is essentially how k-nearest neighbor as a well-established method works: assign a test example the mean value (or majority label) of its k-nearest neighboring train examples. One factor that tunes the granularity (sort of like stratified sampling in statistics) to mitigate the issue of crudeness is the value of k, which is what we used in this study. We demonstrate in this work is that even this crude approach can already lead to improved performance. The transfer learning that we explore here has potential to usher in a new paradigm on how learning can be done. As future research, we will undoubtedly pursue to devise better and more sophisticated ways to transfer atomic features. And actually publishing our work in progress is very likely to inspire more people to work on this problem and find better solutions.

Comments on the Quality of English Language

The author's entire manuscript is riddled with formatting and word errors, with only a portion listed below. These issues reflect a lack of seriousness during the manuscript preparation stage.

 Response: We truly appreciate the reviewer’s comments, and have proof read the manuscript carefully and fixed format and language issues as best as we can. 

line 75 "model,AT-TSVM", should be "model, AT-TSVM,"

fixed.

line 79 "Figure Figure 1.c" should be "Figure 1.c"

fixed.

line 151 "zero otherwise Then," should be "zero otherwise. Then,"

fixed.

line 196 "The Euclidian distanced(p,q)" should be "The Euclidian distance d(p,q)"

fixed.

line 177 "Figure.3(b) llustrates" should be "Figure.3(b) illustrates"

fixed.

Reviewer 3 Report

Comments and Suggestions for Authors

The authors presented a very interesting and novel approach to predict inter residue contacts for transmembrane (TM) helical proteins. This is a very important tools for protein structure prediction and drug discovery as many of the TM proteins are major drug targets. While the manuscript presents an interesting machine learning approach, I believe it is currently written primarily from a computer science perspective. As it stands, it lacks sufficient biological context and validation to make it fully accessible and useful to the structural biology and bioinformatics communities. Additional studies and more biologically grounded evaluations are needed to make the work more impactful and complete.

  1. There is no comparison with the ground truth for the test sets. The authors should report how accurate these predictions are when compared the contact map/ distances calculated from the 3D structure. I would also recommend the authors compare this method with other state-of-the-art methods for inter residues distance prediction. For this comparison, the authors may choose only sequence-based methods as the structure-based methods are not practical as the authors mentioned.
  2. The actual distances are more valuable than the binary contacts to build the structure. Is predicting distances are a more difficult task in this context?
  3. There should be a discussion on how AlphaFold3 performs on TM helical protein structure prediction. There have been some study using AF2, and the results are quite promising. I would recommend that authors predict AF3 structures for these test targets, report the average score, (and trace back the inter residue distances) and compare how that compares with AT-TSVM method.
  4. The manuscript does not mention what distance cutoffs are used for the classification labels. Tools like GREMLIN predict similar contact maps from the evolutionary information. So, the performances should be compared.
  5. The model uses training and testing data together during the training stage. So, it is hard to gauge how transferable this method would be for other transmembrane helical proteins outside the DeepHelicon database. The authors should provide this test also.
  6. There should be some discussion on what features turn out to most important in this classification model. This can be important from the structural biology point of view.
  7. I would recommend the authors to share the complete code in the github during the revision of manuscript.
  8. Figure 1 should be replaced with one with a better resolution and larger font size.

Author Response

Reviewer#3

The authors presented a very interesting and novel approach to predict inter residue contacts for transmembrane (TM) helical proteins. This is a very important tools for protein structure prediction and drug discovery as many of the TM proteins are major drug targets. While the manuscript presents an interesting machine learning approach, I believe it is currently written primarily from a computer science perspective. As it stands, it lacks sufficient biological context and validation to make it fully accessible and useful to the structural biology and bioinformatics communities. Additional studies and more biologically grounded evaluations are needed to make the work more impactful and complete.

  1. There is no comparison with the ground truth for the test sets. The authors should report how accurate these predictions are when compared the contact map/ distances calculated from the 3D structure. I would also recommend the authors compare this method with other state-of-the-art methods for inter residues distance prediction. For this comparison, the authors may choose only sequence-based methods as the structure-based methods are not practical as the authors mentioned.

Response: The ground truth labels for residue pair contact are actually obtained from the 3D structure, as detailed in DeepHelicon from which we adopted the dataset: a contact point is defined as 2 residues that are separated by a minimum of 5 residues in sequence and for which the minimum distance between any pair of their heavy atoms measures less than 5.5Å.  In revision, we added more info regarding how the ground truth labels are established. DeepHelicon, winning out a bunch of the state-of-the-art methods, such as Gremin, MetaPSICOV and DeepMetaPSICOV. Therefore, in a sense we only compare with DeepHelicon. We addressed the comparison issue, in the response to reviewer 1’s comments above, with a caveat that it may not be a fair comparison due to the different settings and learning modes (inductive vs transductive). 

2. The actual distances are more valuable than the binary contacts to build the structure. Is predicting distances are a more difficult task in this context?

Response: The reviewer is right in pointing out that distances are more useful to build the 3D structure. That task, called distance map prediction in literature, is indeed much more difficult than predicting the binary contact map; the latter is where our work belongs to, like DeepHelicon and a whole host of other methods that just predict contact map, not distance map. Contact map, other than as an intermediate step to aid 3D structure prediction, is useful in its own right; for example, knowing whether two residues contact (i.e., forming a covalence bond) at critical position in the sequence such as binding site can help better understand the protein’s function, without having an accurate structure of the whole molecule.

3. There should be a discussion on how AlphaFold3 performs on TM helical protein structure prediction. There have been some study using AF2, and the results are quite promising. I would recommend that authors predict AF3 structures for these test targets, report the average score, (and trace back the inter residue distances) and compare how that compares with AT-TSVM method.

Response: While we really appreciate the reviewer’s suggestion, using AF2 and AF3 really puts the whole study under a different premise and beyond its current scope. AF2 and AF3 are for 3D structure prediction, with great success. What the reviewer suggested is amount to treating their predicted structure as a sort of Oracle (or surrogate to the ground truth) in an active learning manner. In the response to the reviewer 1 above, we already mentioned that we limited inquiries to a minimal number of test targets and then excluded them from evaluating the performance to avoid cheating. The contributions from our study have two folds; in addition to improve the residue contact prediction, it is also exploring a new transfer learning paradigm – features that are useful but only available for training examples.  As the reviewer suggested, words in this effect are added to the discussion section in the revised manuscript to provide adequate perspectives to the reader.

4. The manuscript does not mention what distance cutoffs are used for the classification labels. Tools like GREMLIN predict similar contact maps from the evolutionary information. So, the performances should be compared.

Response: As mentioned in response to the reviewer’s comment #1 above, DeepHelicon, winning out a bunch of the state-of-the-art methods, such as Gremlin, MetaPSICOV and DeepMetaPSICOV. Therefore, in a sense of transitivity, we only compare with DeepHelicon. In revised manuscript, those methods are cited in the reference to the reader adequate background.

5. The model uses training and testing data together during the training stage. So, it is hard to gauge how transferable this method would be for other transmembrane helical proteins outside the DeepHelicon database. The authors should provide this test also.

Response: This seems to be related to the comments from reviewer 2, with respect to “lack of distribution guarantee”. For our method and any other machine learning and statistical learning methods to work, the premise is that training examples and testing examples are drawn i.i.d. from the same data source. Under this premise, the method should work for other transmembrane helical proteins outside the DeepHelicon database, as long as they belong to the same overall data source. Otherwise, the models trained on certain data cannot generally be expected to work well on data that are from a difference source with a different underlying distribution. That said, our method still can be used for the “other transmembrane helical proteins outside the DeepHelicon database”; we just need some examples from these other proteins with known residue contacts to serve as training example. Actually, one advantage of our method is that we need a small training set, because the test examples can participate in training as unlabeled data, in the transductive learning mode.

6. There should be some discussion on what features turn out to most important in this classification model. This can be important from the structural biology point of view.

Response: There are two type of features used in our study: sequential features and atomic features. For sequential features, the co-evolutionary features are most informative, as the study in the reference [14] shows.  Our contribution in this work is to explore utility of atomic features: mean distance between atoms, relative angle between amino acid and inter-helical angle, which have much clear and intuitive meanings in comparison to that of the sequential features. Among these atomic features, the most informative is the mean distance, as reported in reference [15]. We added more info regarding feature’s information in the revision.  

7. I would recommend the authors to share the complete code in the github during the revision of manuscript.

Response: Thanks for the suggestion. A github page is set up for this project: https://github.com/Bander-Almalki/AT-TSVM/   where all code for the work is deposited, though more documentations are still being added to make the package easy to be deployed by the users.

8. Figure 1 should be replaced with one with a better resolution and larger font size.

Response: Thanks for bringing this to our attention. We will work with the journal’s production team to reproduce the figure to their required resolution, if our revised manuscript is kindly accepted.

Round 2

Reviewer 1 Report

Comments and Suggestions for Authors

The reviewer acknowledges the updates to the manuscript and does not have further comments.

Author Response

The authors are grateful that the reviewer accepts their responses.

Reviewer 2 Report

Comments and Suggestions for Authors

The authors have provided corresponding responses to all other questions raised by the reviewer.

In lines 307 to 325, the authors have clearly articulated the practical intention of active learning in this work (though the expression needs to be more academic), which addresses the reviewer's question.

There are certain aspects that still require further improvement.

01. The capitalization in "Figure 4. Updating Test Examples’ Atomic Features using KNN" should be made consistent.

02. The terminology should be unified throughout the response - "testing examples" and "test examples" should be standardized.

03. The resolution of Figure 1 requires enhancement, and the aspect ratio of the text labels needs further adjustment.

Author Response

The authors are grateful that the reviewer accepts their responses to the first round of comments. Below are the responses to the second round of comments. All changes are highlighted in the revised manuscript.

In lines 307 to 325, the authors have clearly articulated the practical intention of active learning in this work (though the expression needs to be more academic), which addresses the reviewer's question.

Response: The expression is rephrased as suggested.

There are certain aspects that still require further improvement.

01. The capitalization in "Figure 4. Updating Test Examples’ Atomic Features using KNN" should be made consistent.

Response: Fixed. Capitalization applies to the first letter of all words in the title.

02. The terminology should be unified throughout the response - "testing examples" and "test examples" should be standardized.

Response: In revision, we standardized as “test examples” throughout the manuscript.

03. The resolution of Figure 1 requires enhancement, and the aspect ratio of the text labels needs further adjustment.

Response: Figure 1 is enhanced, larger fonts are used for text labels, which are more legible now. 

Reviewer 3 Report

Comments and Suggestions for Authors

The authors have addressed all my comments. I recommend publishing the manuscript.

Author Response

The authors are truly grateful that the reviewer accepts their responses and recommends for publication.